# Mechanisms Involved in the Link between Depression, Antidepressant Treatment, and Associated Weight Change

**DOI:** 10.3390/ijms25084511

**Published:** 2024-04-20

**Authors:** Tomas Kukucka, Nikola Ferencova, Zuzana Visnovcova, Igor Ondrejka, Igor Hrtanek, Veronika Kovacova, Andrea Macejova, Zuzana Mlyncekova, Ingrid Tonhajzerova

**Affiliations:** 1Clinic of Psychiatry, Jessenius Faculty of Medicine in Martin, Comenius University in Bratislava, University Hospital Martin, 03659 Martin, Slovakia; kukucka17@uniba.sk (T.K.); igor.ondrejka@uniba.sk (I.O.); igor.hrtanek@uniba.sk (I.H.); kovacova400@uniba.sk (V.K.); macejova5@uniba.sk (A.M.); mlyncekova3@uniba.sk (Z.M.); 2Biomedical Centre Martin, Jessenius Faculty of Medicine in Martin, Comenius University in Bratislava, 03601 Martin, Slovakia; nikola.ferencova@uniba.sk (N.F.); zuzana.visnovcova@uniba.sk (Z.V.); 3Department of Physiology, Jessenius Faculty of Medicine in Martin, Comenius University in Bratislava, 03601 Martin, Slovakia

**Keywords:** depressive disorder, weight gain, antidepressant treatment, food intake regulation, appetite

## Abstract

Major depressive disorder is a severe mood disorder associated with a marked decrease in quality of life and social functioning, accompanied by a risk of suicidal behavior. Therefore, seeking out and adhering to effective treatment is of great personal and society-wide importance. Weight changes associated with antidepressant therapy are often cited as the reason for treatment withdrawal and thus are an important topic of interest. There indeed exists a significant mechanistic overlap between depression, antidepressant treatment, and the regulation of appetite and body weight. The suggested pathomechanisms include the abnormal functioning of the homeostatic (mostly humoral) and hedonic (mostly dopaminergic) circuits of appetite regulation, as well as causing neuromorphological and neurophysiological changes underlying the development of depressive disorder. However, this issue is still extensively discussed. This review aims to summarize mechanisms linked to depression and antidepressant therapy in the context of weight change.

## 1. Introduction

Major depressive disorder (MDD) represents a mood disorder characterized by pervasive low mood, loss of pleasure or interest in enjoyable activities, low self-esteem up to the point of suicidal ideation, psychomotor retardation/agitation, excess fatigue, cognitive difficulties, and a disruption of normal sleep and dietary patterns. Apart from MDD, the depressive disorders category as part of The Diagnostic and Statistical Manual of Mental Disorders, Fifth Edition (DSM-5) also includes disruptive mood dysregulation disorder, premenstrual dysphoric disorder, substance-induced depressive disorder, and persistent depressive disorder, also known as dysthymia. MDD does not explicitly feature in the 10th revision of the International Statistical Classification of Diseases and Related Health Problems (ICD-10); however, its diagnostic criteria closely resemble those of depressive episodes (mild, moderate, severe, or psychotic). However, ICD-11, effective from 2019, aligned the depressive criteria for depressive episodes more closely with DSM-5, most notably including the psychomotor agitation/retardation symptom [1].

Several studies estimating the prevalence of depressive disorders display a significant heterogeneity in results, ranging from 3.44% [2] to 17.3% [3] during the pre-pandemic period. As for MDD itself, the European Health Interview Survey (EHIS) and its further re-evaluation estimates the 2014 European MDD prevalence rate at 2.1% [4]. Up to half of patients having once suffered from MDD will experience a relapse of the disorder within their lifetime. Notably, MDD may cause significant and long-term decreases in an individual’s educational, marital, and career performance [5]; up to 31% of patients suffering from MDD attempt suicide [6]. However, following the COVID-19 pandemic outbreak, a clear increasing trend has been observed. Although the inter-study heterogeneity remains, a 2020 pooled prevalence of depressive disorders has been set at 25% [7]. Thus, the burden of depression is becoming a serious society-wide problem.

The primary method of diagnosing MDD is the semi-structured interview, a core part of the psychiatrist’s examination. Moreover, standardized and validated questionnaire-based methods tend to skew the false positivity rates in favor of overdiagnosis and are thus better suited for screening and research purposes [4]. In contrast to somatic diseases, mental disorders often lack relevant and/or reliable laboratory diagnostic methods. Currently, panels of various parameters related to MDD-related pathophysiological mechanisms (such as α-1 antitrypsin, apolipoprotein CIII, BDNF, cortisol, epidermal growth factor, myeloperoxidase, prolactin, resistin, and soluble tumor necrosis factor α type II) are being investigated [8]. In this context, objective methods such as magnetic resonance imaging (MRI) or positron emission tomography (PET) play a differential diagnostic role, used mainly to rule out potential organic causes of MDD-like symptoms, e.g., in the case of dementia or focal epilepsy. More specifically, several studies using MRI have observed cytoarchitectural changes in the limbic–cortical circuits of depressed patients [9,10] and studies using PET consistently report increases in glucose metabolism in the amygdala as the brain center responsible for negative affects within the context of decision making and memory. Additionally, localized reductions in blood flow and metabolism correlating with the symptoms of psychomotor retardation have also been observed among depressed patients [9,11]. Due to a lack of reverse inference (the inability to deduce clinical symptoms based on a brain scan), neither MRI nor PET are currently included in the diagnostic criteria for MDD [10].

MDD itself has the potential to affect body weight by causing a reduction in physical activity as well as inducing dietary changes. The typical ‘melancholic’ form of MDD is accompanied by decreased appetite and weight loss. In contrast, the atypical form of MDD is accompanied by an increase in appetite and weight gain. The nature of this impact on appetite (increase or decrease) remains the same across multiple depressive episodes in approximately 80% of cases [12] and there appears to be a major overlap between appetite dysregulation among depressed and obese patients but non-depressed patients. Indeed, the risk of developing obesity in depressed patients is 58% higher than in non-depressed subjects [12,13,14]. On the other hand, obesity itself can lead to the development of depression by negatively affecting self-image and by potentiating pro-inflammatory activity. The risk of developing depression in an obese patient is 55% higher than in a lean subject and BMI has been shown to correlate with the intensity of depressive episodes [14,15]. Importantly, chronic stress, another condition related to adiposity changes, is associated with changes in systems affected by depression as well: beta-adrenergic, glucocorticoid, parasympathetic, and NPY-mediated signaling [16].

Weight gain, as an adverse effect of psychopharmacotherapy, is a well-known phenomenon experienced by over 65% of long-term antidepressant users and is one of the leading reasons for spontaneous withdrawal from treatment [17]. A potential way to mitigate such non-adherence may be found in personalized prescriptions of antidepressants according to the patient’s pre-existing metabolic condition. However, the mechanisms linking antidepressant treatment to weight changes are still a matter of discussion and more comprehensive research is needed before a clear model of the relationship can be established. While several psychopharmaceutical drugs (such as tricyclic antidepressants (TCAs) or second-generation antipsychotics) have been known to cause weight gain, others (such as the NDRI bupropion) display a consistent weight-reducing effect or appear weight-neutral (such as SSRIs and SNRIs) [17,18,19]. A noticeable trend exists in the prescription of antidepressants of moving toward newer less weight-affecting molecules [20]. However, the mechanisms linking depression, antidepressant treatment, and weight control are still unclear. Therefore, this review aims to highlight the neurophysiological and molecular mechanisms linking depression and antidepressant therapy effects on weight change.

## 2. Physiology of Appetite Regulation: The Homeostatic and Hedonic Systems

In order to discuss the weight fluctuations attributed to depression and antidepressant therapy, it is first necessary to discuss the intricate systems governing appetite. Appetite, the drive to consume food items, develops as an expression of the human body’s regulatory processes serving to maintain its energetic and metabolic needs. It consists of an episodic component leading to individual meals and a tonic component representing the global energy requirement of the body. The tonic component is modulated mostly by hormonal regulation, particularly by the activity of orexigenic and anorexigenic hormones, linked primarily to the ‘homeostatic system’. Furthermore, the episodic component is modulated mainly by serotonergic and dopaminergic neurotransmission, linked foremostly to the ‘hedonic system’. Both components ultimately sum up in hypothalamic circuits [21].

### 2.1. The Homeostatic System: Regulation Based on Energy Demand

The homeostatic system of food intake regulation consists of peripheral endocrine and metabolic signals acting upon specific areas of the hypothalamus, parabrachial nucleus, and the nucleus of the solitary tract. It is responsible for maintaining a basal energetic intake in accordance with the needs of the organism [13,22,23].

Agouti-related protein (**AgRP**) is one of the most important orexigenic neuropeptides and is considered the final biochemical representation of hunger, integrating the afferent signaling via neuropeptide Y (NPY), leptin, GABAergic neurons, and others. Increases in AgRP levels lead to the preference of food-seeking behaviors over other competing motivated behaviors and thus, the initiation of feeding. AgRP neuronal activity has recently been associated with the development of depressive symptoms, potentially expanding their role as mere appetite regulators [24,25]. **NPY**, secreted intracerebrally and likewise from peripheral sympathetic neurons, has a hyperphagic and anxiolytic impact and induces adipose tissue proliferation in the periphery. Stress-induced adipose tissue remodeling is thus mediated to a significant extent by NPY. Its signaling inversely correlates with hypothalamic serotoninergic neurotransmission and has been inversely associated with symptom severity of MDD and blood levels of corticotropin-releasing hormone (CRH) [16,21,26]. **Ghrelin**, secreted in the stomach, is thought to be a functional antagonist of leptin. It has an orexigenic effect, mainly elucidating the intake of fat-rich foods, and acts via potentiating the influence of NPY and AgRP and also via inhibition of the pro-opiomelanocortin (POMC) system [27]. It interacts with both the homeostatic and hedonic circuits and also leads to the uptake of fatty acids by adipocytes. Ghrelin levels are lower among obese individuals than in the healthy population—probably due to an adaptation to a long-term positive energy balance. Weight loss is also correlated with an increase in ghrelin levels. Moreover, the role of ghrelin is also implied in the regulation of sleep and memory function whose impairments are associated with depressive disorders. An association has been observed between the hyper-/hypophagic variant of MDD and hedonic system activity in response to post-prandial ghrelin levels increase, whereby in hyperphagic variants of MDD hedonic system activity rises and, vice versa, hypophagic variants of MDD lead to a hedonic system activity decrease [28]. Ghrelin levels are known to decline with antidepressant treatment. However, the effect of depression on ghrelin levels is not yet clearly established [29].

**Leptin** is a peripheral hormone acting as a feedback signal informing on the state of adipose tissue fat stores. It has an anorexigenic effect through inhibition of NPY and AgRP and activation of the POMC system and cholecystokinin signalization [27,30]. Its most significant site of action is the arcuate nucleus but it also indirectly attenuates mesolimbic dopaminergic activity via hypothalamic orexin neurons projecting to the ventral tegmental area (VTA) [31]. Its serum levels correspond with body fat levels but obese populations have chronically increased appetite even with high levels of circulating leptin. The catecholamine signaling pathway through β2,3 receptors inhibits leptin synthesis. In women, leptin levels are higher than in men and this difference is significantly more pronounced in women suffering from depressive disorders. This has been attributed to depression-linked increases in cortisol levels resulting in increased stimulation of leptin secretion [32]. Increased leptin levels have been correlated to disordered eating (emotional eating, deliberately restrained eating, and increased eating in response to external sensory cues) in association with MDD [33]. **POMC** is an anorexigenic neuropeptide acting via the stimulation of mineralocorticoid receptors in the lateral hypothalamus. Moreover, it exerts an inhibitory influence upon mesolimbic dopaminergic signaling, therefore potentially decreasing hedonic drive. It is considered to be the biochemical representation of the satiety sensation, leading to the cessation of feeding activity [30,34,35]. **Peptide YY** (PYY) is an anorexigenic peptide secreted postprandially in the small intestine known for its modulatory influence on VTA activity. Its influence leads to a shift in the regulation of food intake from the homeostatic circuit to the hedonic circuit. Unlike leptin, obese patients do not develop resistance to PYY [36,37,38]. **Cholecystokinin** is an anorexigenic peptide secreted in the small intestine, whose biological function includes the regulation of gastric and biliary emptying. In addition to vagal neurotransmission to the homeostatic circuit, it also induces activity in the hedonic circuit. Anorexigenic peptides act on receptors in vagal afferents. The rate of their secretion also partly depends on the concentration and character of fatty acids in one‘s diet [23,27]. 

### 2.2. The Hedonic System: Regulation Based on Reward

The mesolimbic dopaminergic system, also known as the hedonic circuit, is a network of brain structures related to rewarding experience, motivation, decision making, and learning. Its influence on the regulation of food intake operates through the positive emotional response to palatable food. It consists mainly of dopaminergic connections between the VTA and the nucleus accumbens (NAC), the amygdala, the prefrontal cortex (PFC), and the hippocampus. Dopaminergic projections to the lateral hypothalamic areas are thought to be the most important connections, due to it being considered the definitive site of appetite summation [13,23]. The PFC is responsible for decision making, behavioral flexibility, and learning of operantly conditioned actions [39]. The medial regions of the NAC and the posterior regions of the ventral pallidum form the neurobiological components of the psychological phenomenon of “liking”. The perception of palatable food, along with other reward-related cues, leads to a dopamine-mediated override of the homeostatic system and hedonically regulated feeding behavior [40].

High-fat food intake elicits a dopamine-releasing effect independent of portion size [41]. The notable shift away from the homeostatic regulatory system in favor of a motivation-based hedonic system is strongly associated with the development of obesity via the phenomenon of food addiction. People suffering from obesity, as with addictions, tend to exceed the originally intended food portions (in the same manner addicts exceed their intended drug intake) and experience a reduction in the range of social and occupational activities undertaken. There is also a phenomenon whereby obese people fail to cease overeating despite being fully aware of the negative impact on their health [13]. Obesity is also associated with changes in the areas responsible for processing gustatory sensory stimuli. Hypoactivity of the reward system likely plays a role in the development of obesity and binge eating may represent a relevant compensatory mechanism, providing the desired feeling of a rewarding dopamine rush. Obese individuals are willing to overcome greater obstacles in an attempt to procure food than lean individuals. There is also an inverse association between dopaminergic receptor availability and BMI [13,23,42]. Indeed, palatable food consumption carries a positive emotional charge and is thus also considered as an anxiety-reducing coping strategy. Among depressed individuals, feeding behavior tends to become more impulsive. McNaugton et al. proposed a role of impaired insular cortex function leading to a misinterpretation of negative internal sensations as hunger in depressed individuals [32].

The μ-opioid receptors of the endocannabinoid system also play an important role in liking. Taking naloxone (an opioid receptor antagonist) leads to a reduced intake of palatable high-calorie foods [13,23,43]. **CB1** cannabinoid receptors downregulate anorexigenic peptides such as cocaine and amphetamine-regulated transcript (CART) or CRH while upregulating the orexigenic NPY [44]. **CART** is an anorexigenic neuropeptide involved in the effects of psychostimulant drugs such as feeding suppression and the experience of being “high”. It exerts a disinhibiting effect on mesolimbic dopaminergic neurons, taking part in the mediation of reward and reinforced learning [45,46]. **Orexin** (also known as hypocretin) regulates food intake and the sleep cycle. Orexin receptors Ox1 and Ox2 are involved in the response to addictive stimuli. Orexin neurons are mainly located in the lateral hypothalamus, from where they project mainly to the dorsal raphe nuclei (serotoninergically), locus coeruleus (noradrenergically), and the VTA (dopaminergically). The activity of the orexinergic system is regulated to a significant extent by serotoninergic signaling, via 5-HT_1A_ and 5-HT_2C_ receptors [21,47]. The mechanisms of both homeostatic and hedonic systems, as well as their relevant regulatory inputs, are summarized in Figure 1.

## 3. Mechanisms Linking Depression and Appetite Regulation

Appetite disturbance and the accompanied weight change are included among the diagnostic criteria of major depressive disorder. From a clinical standpoint, a distinction is to be made between the typical ‘melancholic’ form of depression whereby food intake is reduced and atypical depression linked with comfort eating and an overall increase in appetite [48]. Simmons et al. [12] link appetite-decreased depression with a state of hypercortisolemia and dopaminergic hypoactivity and appetite-increased depression with increased inflammatory and dopaminergic response activity to food cues. Both types of depression are considered states of chronic psychosocial stress. Acute or short-term stress leads to an intensification of dopaminergic signaling in the NAC, which leads to more pronounced reward-seeking activities alongside the increased learning of goal-achieving behavior—a constructive and effective way of coping with stressors [49]. On the other hand, chronic or long-term stress leads to a decrease in dopaminergic signaling in the NAC and VTA hypoactivity. Moreover, chronic stress leads to a persistent state of physiological arousal, which in turn utilizes its own set of biological and behavioral adaptations. Depression is associated with hypersensitivity to negative stimuli and therefore leads to an aggravated influence of previously sub-threshold stressors [50]. The chronic stress adaptation mechanisms involve the activation of the hypothalamic–pituitary–adrenocortical (HPA) axis. This leads to an increased CRH secretion in the paraventricular nucleus of the hypothalamus and consequently an increase in blood cortisol levels [51,52,53]. Thus, depression is characterized by a state of hypercortisolemia. Due to cortisol’s role in appetite and metabolism regulation as well as adipocyte differentiation, hypercortisolemia and HPA axis hypersensitivity are known risk factors in the development of obesity, considered even more potent than a high-fat diet [54,55,56]. Thus, a vicious circle emerges: chronic stress associated with long-term hypercortisolemia favors the development of obesity and also increases the odds of developing depression—a state of increased chronic stress.

In the context of cortisol levels and depression, an important hypercortisolemia effect to discuss is the remodeling of mesolimbic dopaminergic neurons. Receptors facilitating dopaminergic signaling in the mesolimbic pathways are divided into **D1-like** and **D2-like** subtypes and, according to their representation, the nucleus accumbens medium spiny neurons (MSNs) are divided into D1 MSNs and D2 MSNs. D1 MSNs express D1, D5, α1, NMDA, AMPA, CB1, and M1,4 receptors and facilitate the sensation of “reward”, with their activity exerting a behavior-intensifying influence. D2 MSNs, on the other hand, exert a behavior-inhibiting influence, facilitating the sensation of “aversion”. They express D2,3,4, α2, NMDA, AMPA, CB1, and M1,4 receptors. Under the influence of chronic stress, MSN dendritic remodeling occurs, which alters the signaling ratio in favor of D2 MSNs. The disparity between their activity is associated with the development of pathological behavioral patterns such as social aversion and anhedonia [49,57,58,59,60]. These depressive behavioral patterns conditioned by repeated stressful stimuli are in animal models dependent on the functionality of glucocorticoid receptors—thus establishing the role of cortisol in depressive behavioral changes [57,61,62]. A noteworthy distinction between “susceptible” and “resilient” individuals has been observed in animal experiments, facilitated by a compensatory upregulation of a corrective K+ current balancing out stress-induced dopaminergic hyperactivity [63].

Neuronal rearrangement of the mesolimbic pathway is associated with the upregulation of **BDNF** (brain-derived neurotrophic factor) mRNA in the NAC. However, stress-related hypercortisolemia leads to reduced BDNF expression and neuronal growth dysregulation. Indeed, according to the neurotrophic hypothesis [64], depression is associated with a disruption in the normal function of neurotrophins—brain growth factors. The neurotrophic hypothesis explains the evidence of the association of depression with neurochemical and structural changes. Thus, BDNF dysregulation also inhibits the ability of glial cells to facilitate synaptic cleft emptying, which may lead to excitotoxicity indicating a state of intrasynaptically released glutamate levels reaching potentially neurotoxic concentrations [65,66,67]. Subsequent structural changes in the CNS neuronal network may explain the excitatory activity changes typical for depression. Indeed, BDNF exerts a region-specific effect, with mainly its signaling cascade between the VTA and the NAC being involved in causing depressive behavior. However, BDNF exhibits antidepressant effects in other central structures such as the hippocampal circuit [43,64].

One of the brain regions prone to depression-related **atrophy** is the anterior cingulate cortex (ACC). It is involved in the regulation of error/mistake detection, attention targeting, impulsivity, decision making, and emotions. Atrophy-based ACC hypofunction leads to autonomic dysregulation commonly seen in depression and to the affective disturbances manifested as anhedonia, apathy, and abulia. Notably, lesions leading to ACC hypofunction led to a reduction in food acquisition in the animal model, particularly in competitive conditions as in the food being “guarded”. However, the appetite itself remains unchanged [68,69,70]. An association is also observed between cortisol levels and activity in the basolateral nucleus of the amygdala, the center responsible for (mainly negative) affective modulation of memory storage. An inverse association of cortisol levels also exists with the excitability of the hippocampus—the center responsible for learning and memory processes. Chronic hypercortisolemia indeed leads to its partial atrophy [66,67]. 

**Noradrenergic** neurotransmission from the locus coeruleus to the dopaminergic neurons of the VTA, a hallmark feature of chronic stress, regulates noradrenergic excitability reflected in the activity of the terminal VTA projecting regions (i.e., NAC and PFC). Repeated stimulation of contributing α1 and β3 receptors leads to increased resistance to hyperactive signaling originating in the NAC [43,59]. Stress also alters excitatory neurotransmission from areas of the thalamus, hippocampus, and PFC [43,49,52]. Stress also leads to **cholinergic** dysfunction. More specifically, the acetylcholine signaling represented by M5 muscarinic receptors modulates VTA activity by increasing the excitability of dopaminergic neurons. Disruption of cholinergic signaling in NAC interneurons leads to anhedonia and also broader cognitive impairment [43,71]. At this point, chronic stress leads to changes in **glutamatergic** neurotransmission in the NAC. The rate of glutamatergic neurotransmission from intralaminar regions of the thalamus to the NAC correlates with social avoidance [43].

Stress also leads to the release of the neuropeptide **dynorphin**, an endogenous kappa opioid receptor (KOR) agonist. KOR receptors are found in the NAC and the VTA, where they lead to a marked inhibition of dopaminergic signaling. Under physiological conditions, this mechanism occurs within the feedback-based autoregulation of the reward system; however, under stress situations, this mechanism leads to a decrease in pleasure signaling and the development of an aversive effect. KOR antagonization has an anorexigenic effect in obese individuals, particularly observable in the reduction of high-caloric food intake [41]. **Enkephalin**, an endogenous opioid agonizing delta opioid receptor, is in turn associated with stress resistance and antidepressive properties [72]. 

According to the cytokine theory of depression suggesting a chronic **proinflammatory state**, the metabolism of tryptophan (a precursor of serotonin and melatonin) is impaired in favor of its increased biotransformation to kynurenine. This leads to a decreased availability of substrate for serotonin synthesis and results in a decrease in serotonergic neurotransmission, a state closely related to depression [73]. Anti-inflammatory treatment has been observed to reduce depressive symptoms and pro-inflammatory cytokines such as interferon alpha are known to aggravate them [74,75]. Chronic stress also disrupts the blood–brain barrier and allows peripheral immunocytes to infiltrate the brain tissue, causing an increase in intracerebral cytokine levels. Perhaps unsurprisingly, obesity as a chronic pro-inflammatory state and is a frequent comorbidity of depression. Hypertrophic adipose tissue creates conditions limiting the possibility to achieve adequate perfusion, leading to an increased occurrence of hypoxia, and thus adipose tissue becomes a source of circulating proinflammatory cytokines. With respect to an inflammatory state, the proinflammatory state also leads to potentiation of the risk of developing depression-associated somatic (e.g., cardiovascular or metabolic) complications. These complex mechanisms can constitute a major link between stress, depression, and metabolic syndrome [13,76], summarized in Figure 2.

## 4. The Effect of Antidepressant Treatment on Weight Change

Depression causes loss of appetite; therefore, it may be difficult to discern appetite increase due to recovery from depression and due to the side-effects of antidepressant medication [77]. Several studies revealed that the severity of depressive symptoms does not seem to correlate well with drug-induced weight change [18]. Compared with antipsychotics, most antidepressants display only a mild effect on body weight. The rate of a medication’s weight-gaining effect during the first month of administration appears to be the best predictor of further weight change trajectory and a ≥5% weight change at this point should prompt a re-evaluation of the treatment approach and/or the incorporation of a weight-control strategy [78]. Moreover, in the case of SSRIs and mirtazapine, the user’s body weight seems to stabilize (albeit still increased compared to pre-treatment) after a period of 24 weeks of treatment [79].

Antidepressant treatment also attenuates elevated blood cortisol levels. This effect has been observed with some SSRIs (selective serotonin reuptake inhibitors, notably citalopram), TCAs (tricyclic antidepressants), and mirtazapine. Moreover, antidepressant treatment has also been shown to correct reduced levels of circulating BDNF. BDNF-lowering effects have been observed with antidepressants from the SSRI class, NRIs (noradrenaline reuptake inhibitors), SNRIs (serotonin and noradrenaline reuptake inhibitors), MAOIs (monoamine oxidase inhibitors), atypical antidepressants, ketamine, lithium, atypical antipsychotics, and with electroconvulsive therapy [64,80]. The degree of BDNF level normalization correlates significantly with the degree of depressive symptom alleviation [81].

With respect to antidepressant medication, the role of intestinal microbiota should not be overlooked. Current research indicates a correlation between depressive symptoms and certain populations of intestinal microbiota (more specifically the genera *Coprococcus*, *Eggerthella*, *Eubacterium ventriosum*, *Hungatella*, *Lachnoclostridium*, *Lachnospiraceae*, *Ruminococcaceae*, *Sellimonas*, and *Subdoligranulum*) [82]. Furthermore, the administration of certain probiotic formulations has also been shown to alleviate depressive symptoms. One’s response to antidepressant medication may thus depend on the composition of their gut microbiome, indicating a link between specific microbe species and MDD treatment resistance [83]. The composition of the gut microbiome also affects intestinal permeability and intra-intestinal drug metabolism (e.g., via bacterial metabolic pathways), altering the pharmacokinetics of any peroral antidepressants. Based on these findings, future studies including “psychobiotics” (psychoactive probiotics) as an emergent antidepressant therapy can help elucidate the pathway between the gut microbiome, antidepressants, and obesity.

### 4.1. Tricyclic Antidepressants

Tricyclic antidepressants (TCAs) represent a classic first-generation formula of antidepressant drugs—the monoamine theory [84]. The use of all TCAs is associated with weight gain, more significantly than other antidepressant classes, due to their less selective binding profile [17,85]. Their mechanism of action depends on the inhibition of serotonin, norepinephrine, and dopamine reuptake, thereby increasing the activity of their respective pathways. 

The serotonin system represents the predominant target in antidepressant treatment. An increase in serotoninergic transmission has the acute consequence of attenuating impulsivity, reward-seeking behavior, and the associated overconsumption of energy-dense foods. However, it does not lead to an impairment of reward-based learning [86]. Stimulation of the **5-HT_2A_** receptors leads to satiety and its inhibition results in overeating and weight gain. In an animal model, its blockade also led to manifestations of insulin resistance: reduced glucose muscle intake, increased hepatic glucose production, and increased cortisol secretion. **5-HT_2C_** receptors play a role in the regulation of substance- and food-motivated behavior. Their stimulation leads to a decrease in impulsive and reward-seeking behaviors and their antagonism induces hyperphagic behavior [44,86]. Stimulation of 5-HT_2C_ receptors leads to the activation of POMC-expressing nucleus arcuatus neurons and thus exerts an anorexigenic influence. It also leads to the attenuation of noradrenergic and dopaminergic neurotransmission [77,87]. 5-HT_2C_ agonists have been also shown to increase insulin sensitivity [88]. Unlike most receptor systems, long-term interaction with not only an agonist but also an antagonist leads to a paradoxical downregulation at 5-HT_2C_ and 5-HT_2A_ receptors. This results in a loss of the anorexigenic effect with prolonged pharmacotherapy and, over time, leads to a return to and a possible exceeding of their original body mass. The mechanism responsible for this paradoxical downregulation is thought to be the increased internalization of receptor–antagonist complexes [89]. 5-HT_2C_ antagonism has also been shown to lead to glucose intolerance, as is the case in 5-HT_2A_ receptors [44]. **5-HT_3_** receptors play a role in the regulation of peristaltic and emetic activity. A significant postsynaptic representation of these receptors is also found in the hippocampus and amygdala. Activation of these receptors in the CNS potentiates dopaminergic, cholecystokininergic, and GABAergic neurotransmission. Blockade of 5-HT_3_ receptors antagonizes the anorexigenic effect of methamphetamine [90,91]. Additionally, the **5-HT_1A_** and **5-HT_1B_** receptors are also involved to a lesser extent in the regulation of food intake and reward stimulus seeking, exerting a hypophagic effect [21,86]. 

Norepinephrine exerts a variable effect on food intake and body weight. Stimulation of **α1** and **β2** receptors leads to the inhibition of food intake and stimulation of **α2** receptors has a disinhibitory (orexigenic) effect [92]. Stimulation of α1 adrenergic receptors leads to an increase in glycemia and stimulation of α2 adrenergic receptors leads to a decrease in glycemia. The increase in noradrenergic transmission also has a weight-reducing effect via **β3** adrenoceptors in adipose tissue, which promotes peripheral catabolic processes. Although it is not the dominant mechanism of psychopharmacogenic weight change, the noradrenergic mechanism of action carries a metabolic effect [17,93].

Impaired mesolimbic dopaminergic function leads to a dysfunction of the hedonic system commonly seen among depressed and/or obese individuals. The transmission rate alone is not the only significant parameter though; the ratio of D1 and D2 MSNs and their neuroarchitectural conditions significantly affect hedonic system functioning as well. **D1** receptor activity is an important mechanism of hedonically motivated feeding. Their hyperactivity is observed in obese individuals and their blockade leads to attenuation of the search for palatable calorie-rich food [42]. The blockade of **D2** receptors also leads to a hyperphagic effect and hypersecretion of prolactin, elevated levels of which are associated with higher BMI values. Among obese patients, there is an observed tendency to compensate for decreased D2 receptor sensitivity by increased food seeking as a source of reward [17,44,93]. Blockade of **D3** receptors leads to the reduced expression of CART [94]. The aberrant dopaminergic activity of the VTA can be corrected by antidepressant therapy as well as electroconvulsive therapy.

Alongside TCA’s desirable effect on monoamine systems, they are known as inhibitors of H1 histamine and muscarinic acetylcholine receptors. It is the inhibition of H1 receptors that leads to weight gain as a side effect [17,95]. It is most commonly reported with amitriptyline, whose effect is also thought to be an increase in leptin resistance [96]. **H1** receptor blockade is associated with increased appetite, increased sugar craving, and weight gain. The degree of a drug’s H1 receptor inhibition correlates with the rate of weight gain experienced. Hypofunction of H1 signaling is associated with reduced anorexigenic function of leptin—thus establishing its need for an intact H1 signaling pathway. Histaminergic neurotransmission is thought to be a satiety signal terminating feeding. **H3** receptor antagonization has an anorexigenic effect, probably through a presynaptic inhibitory effect [17,93,97,98]. High antihistaminergic activity is considered the reason why some drugs cause more weight gain than others of the same class. In animal models, **M3** muscarinic receptor dysfunction is associated with lower rates of food consumption and hence lower subject body weights. Cholinergic neurotransmission to the nucleus arcuatus also leads to decreased secretion of POMC, an anorexigenic neuropeptide [99]. Although this hypothesis is a subject of debate, it is also thought that M3 blockade leads to an impaired insulin-secretory response to hyperglycemia [17,93,100].

### 4.2. Selective Serotonin Reuptake Inhibitors

Today, the most widely used antidepressants belong to the selective serotonin reuptake inhibitors (SSRIs) group. In comparison with TCAs, their use exhibits similar efficacy against depressive symptoms with a significantly lower severity of side effects [101]. The use of **SSRIs** often leads to weight loss in the acute phase attributed to a reduction in impulsivity and associated overeating. This phenomenon led to the investigation into some antidepressants’ (mostly SSRIs) potential use as antiobesics [77]. However, long-term (6 months or more) treatment is associated with the re-establishment of pre-treatment weight values and their possible exceeding during the maintenance treatment [76]. Overall, weight gain is reported in 49% of SSRI users [101]. The biphasic effect on weight gain is thought to be caused by eventual 5-HT_2C_ receptor downregulation. Some SSRIs exhibit a higher level of interaction with the histaminergic system, leading to a more pronounced weight-gaining potential, such is the case of the SSRIs paroxetine and citalopram [102,103]. As an important side effect, the early period of SSRI use (especially the first 4 weeks [101]) tends to be accompanied by gastrointestinal disruptions such as dyspepsia, nausea, diarrhea, or constipation. Such peripheral effects of serotonin elevation may provide a secondary explanation of biphasic feeding pattern change, operating outside the central appetite regulatory circuits.

With respect to the anthropometric profile, Calarge et al. [104] tracked the effect of adolescent SSRI use on body composition over 1.5 years and uncovered a positive (weight-increasing) effect on BMI, visceral fat level, and both fatty and lean mass proportionally in adolescent depression. Interestingly, SSRI use during pregnancy is associated with lower infant birth weight and intrauterine growth restriction. An increased need for laxative use in the first five years of life has also been observed in children with prenatal exposure to SSRIs. This suggests an effect of SSRIs on the regulation of the function of the developing gastrointestinal tract. This hypothesis is supported by animal studies that attribute serotonin to a regulatory effect on neuronal growth [105]. The administration of fluoxetine, a commonly prescribed SSRI, increases the sensitivity of OBRb receptors for leptin in a murine model and it also induces BDNF expression [106]. Fluoxetine treatment has also been shown to achieve retrograde modification of stress-conditioned dendritic remodeling of the NAC MSN neurons—thus correcting a neuroarchitectural basis of a dysregulated hedonic system [107].

### 4.3. Serotonin and Norepinephrine Reuptake Inhibitors

Serotonin and norepinephrine reuptake inhibitors (SNRIs) are considered an evolution of previously established SSRI antidepressants, also being able to affect depression-disrupted norepinephrine metabolism (e.g., in the limbic system). Thus, the activating effect of noradrenaline reuptake inhibition enables a more pronounced early mass-negative effect than that of SSRIs alone and, over the longer term, SNRIs are considered to exert a mass-neutral effect [17,108]. Unlike SSRIs, SNRIs also appear to have a more pronounced dose-dependent effect whereby an increase in dosage increases noradrenaline reuptake inhibition [17]. Duloxetine, an SNRI with a more potent serotoninergic action when compared to the rest of the group, is also associated with the most pronounced weight-gaining effect [108,109]. Overall, however, studies examining the effect of SNRIs on body weight are scarce.

### 4.4. Atypical Antidepressants

Bupropion, a noradrenaline and dopamine reuptake inhibitor (**NDRI**), is the only antidepressant with a consistently demonstrated weight loss effect. Bupropion inhibits noradrenergic signaling in the locus coeruleus and increases dopaminergic signaling in the mesolimbic dopaminergic system. It also induces mild psychostimulant effects, which may lead to increased energy expenditure. It has not been shown to have any dopamine-releasing activity. A bupropion–naltrexone combination has been in use as an anti-obesity drug, as the magnitude of achieved loss correlates with the patient’s initial degree of overweight. Recent studies involving bupropion have demonstrated an inhibitory effect on 5-HT_3A_ receptors playing a role in the regulation of peristalsis and emesis, as well as playing a role in the anorexigenic effect of amphetamines [17,19,91,95,110,111].

Mirtazapine, a noradrenergic and specific serotonergic antidepressant (**NaSSA)**, acts via antagonization of 5-HT_2_ and 5-HT_3_ receptors, the enhancement of 5-HT_1_ receptor signalization, and antagonization of α_2_ receptors of the noradrenergic system. Compared to SSRIs, it boasts a more rapid onset of action and, compared to TCAs, it seems to elicit a higher sustained remission rate than amitriptyline [112]. However, as a strong antagonist of H1 receptors and moderate antagonist of muscarinic receptors, it also carries the potential to cause severe weight gain [17]. Blockade of the 5-HT_2A_ receptor also leads to excessive NPY excretion and disinhibition of orexigenic signaling, thereby disrupting appetite regulation [17,93]. 

**Agomelatine** is an antidepressant boasting a novel mechanism of action: MT1/MT2 melatonin receptor agonism, alongside 5-HT_2C_ receptor antagonism. Compared to other antidepressants, it possesses a favorable side-effect profile, with dizziness being reported as the only significant issue. Research on agomelatine’s effect on weight remains scarce but suggests an absence of any significant changes [113]. In murine models, a reduction in melatoninergic signaling has been associated with weight gain and insulin dysfunction. Agomelatine has been shown to lead to decreased fattening and a decreased degree of dyslipidemia among rats on a high-fat diet, suggesting a metabolically protective potential effect of agomelatine use.

**Vortioxetine** is a novel multimodal antidepressant acting through serotonin reuptake inhibition combined with 5-HT_1A,B_ agonism and 5-HT_1D,3,7_ antagonism [114]. It is frequently used as a second line of treatment upon unsatisfactory response to a chosen first-line SSRI [115]. Available research suggests a long-term weight-neutral effect with less than 1 kg gained [115,116,117]. 

### 4.5. MAOIs

Monoamine oxidase inhibitors work by inhibiting the degradation of the psychoactive monoamines dopamine, serotonin, and noradrenaline. Their side-effect profile depends on the degree of selectivity and reversibility of inhibition. Although MAOI class antidepressants will increase intrasynaptic dopamine levels, spontaneous presynaptic activity will be suppressed due to feedback inhibition and thus offer a less significant correction of the dopaminergic neurotransmission [59]. They are associated with only a modest weight-gaining effect, mainly due to increased sugar craving due to an increase in glucose transport leading to hypoglycemic tendencies [95,118]. In addition, phenelzine, a representative of this group, interferes with adipose tissue development itself [119]. 

The effect of the antidepressant therapy on weight is summarized in the Table 1. 

## 5. Conclusions

Weight gain represents an important problem associated with pharmacotherapy in mental disorders. However, the exact mechanisms of antidepressants on weight are still not fully understood. Thus, further research can elucidate novel knowledge linked to the pathomechanisms and the effect of antidepressant treatment on weight abnormalities, which is important for personalized therapeutic management of depressive patients, particularly those at the vulnerable adolescent age.

## Figures and Tables

**Figure 1 ijms-25-04511-f001:**
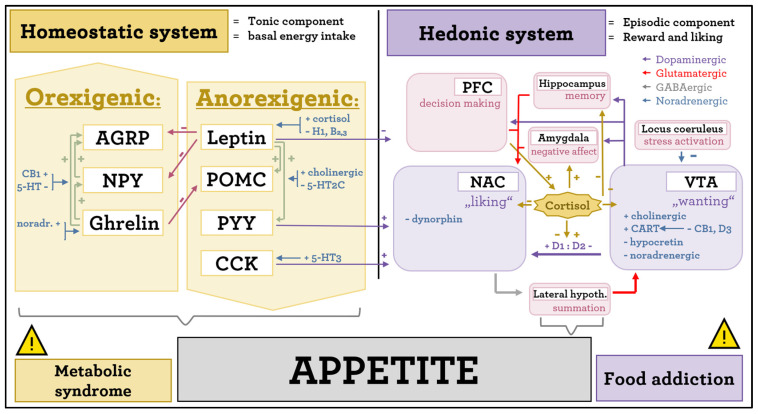
An overview of appetite regulation circuits and relevant regulatory inputs. The homeostatic system operates mostly on a humoral basis, via a relative balance of orexigenic and anorexigenic peptides, with influence on the rate of dopaminergic neurotransmission. Dysregulation of the homeostatic system leads to an increased risk of metabolic syndrome. The hedonic system operates mostly via the mesolimbic reward pathways and is thus sensitive to substances affecting neurotransmission, as well as changes in the D1/D2 MSN neuroarchitecture. Dysregulation of the hedonic system leads to an increased risk of food addiction. Disruption of either system can act as a potential mechanism of obesity development. PFC–prefrontal cortex, NAC–nucleus accumbens, VTA–ventral tegmental area.

**Figure 2 ijms-25-04511-f002:**
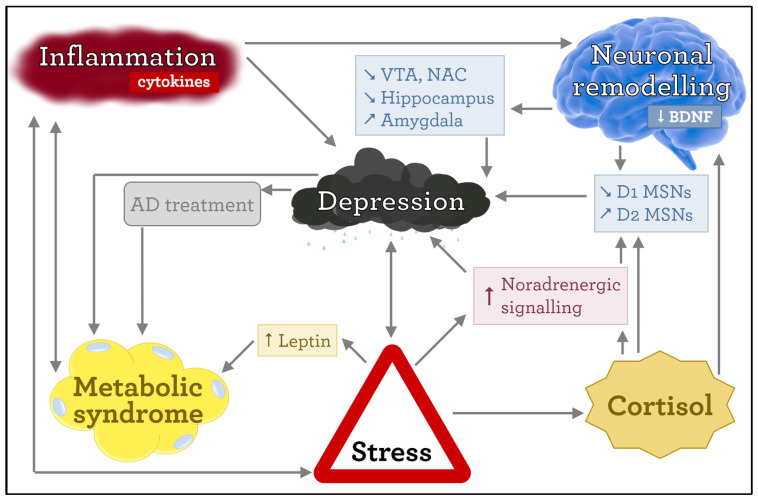
Several pathophysiological mechanisms link stress, depression, and metabolic syndrome. BDNF—brain-derived neurotrophic factor, VTA—ventral tegmental area, NAC—nucleus accumbens, MSNs—medium spiny neurons, AD treatment—antidepressant treatment.

**Table 1 ijms-25-04511-t001:** Common antidepressants and their effect on weight. +/- indicate a stimulating or inhibitory effect on the relevant target and o indicates a negligible effect on weight. Notably, the biphasic effect of certain SSRIs facilitated 5-HT_2C_ downregulation. Abbreviations: SSRI—selective serotonin reuptake inhibitors, SNRI—serotonin and noradrenaline reuptake inhibitors, NDRI—noradrenaline and dopamine reuptake inhibitors, NaSSA—noradrenergic and specific serotonergic antidepressants, MAOI—monoamine oxidase inhibitors, TCA—tricyclical antidepressants.

**Class**	**Drug**	**Effect on Weight**	**Mechanisms Involved**	**Data Sources**
SSRI	Fluoxetine	-, o	5-HT +, 5-HT_2C_ -, leptin +, cortisol -	[17,18,19,95,108]
	Sertraline	-, o	5-HT +, D +	[17,19,108,113]
	Paroxetine	+	5-HT +, M3 -	[17,18,19,102,108]
	Citalopram	+	5-HT +, H1 -	[17,19,102,108]
SNRI	Venlafaxine	-	5-HT +, α1 +	[17,19,108]
	Duloxetine	o	5-HT +, α1 +	[17,19,108]
NDRI	Bupropion	--	α +, D +	[17,18,19,108,110]
Atypical	Mirtazapine	++	H1 -, α2 -, 5-HT -	[17,18,19,96,108]
	Vortioxetine	o	5-HT +/-	[17,115,116,117]
	Agomelatine	o	5-HT +, MT1/2 +	[113]
MAOI	Moclobemide	o	monoamines +	[17,19,118]
	Selegiline	+	monoamines +, glucose -	[84,118,120]
	Phenelzine	+	monoamines +, glucose -	[17,19,118]
TCA	Clomipramine	+	H1 -, M3 -, α1 -	[19,96]
	Imipramine	+	H1 -, M3 -, α1 -	[17,19,108]
	Amitriptyline	++	H1 -, M3 -, α1 -	[17,18,19,96,108]

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
