# Peer review of "Mechanisms Involved in the Link between Depression, Antidepressant Treatment, and Associated Weight Change"

_ijms, 2024, doi:10.3390/ijms25084511_

Round 1
Reviewer 1 Report
Comments and Suggestions for Authors
1. The review study is average but scientifically sound. The Author should include in the Introduction, the following :
-The epidemiology of the disease and related problems in brief
-Classification of depressive disorders according to DSM-IV & 1CD-10 guidelines in brief
-Author to clarify line 34 in the Introduction part of the manuscript ("same in ca."?)-please explain in detail.
2. Techniques in brief for diagnosing the disease and its related outcome -e.g. Brain imaging techniques (MRI and PET scan etc.)
3. In brief about the interplay between stress, Depressive disorder and obesity
4. The Author may also include a Meta-analysis of this study in this review
5. The Author should also clarify whether Figures 1 & 2 are self-made schematic diagrams or taken from any other source and if so then he should mention and give the consent from where it has been taken.
Reviewer 2 Report
Comments and Suggestions for Authors
Gongrats, your review is well written and very well structured which makes it easy to read and understand. However, an important point that I suggest you to consider is the possibility of analyzing the role of intestinal absorption of treatments for depression, as an additional mechanism since, as has been reported, that in overweight and obesity there is a significant alteration in the intestinal barrier affecting absorption
Reviewer 3 Report
Comments and Suggestions for Authors
Ok
Reviewer 4 Report
Comments and Suggestions for Authors The ms is a timely review describing the interplay between the pathophysiology/treatment of depression and appetite regulation/body weight. The text is well written and complete, ranging from the description of the physiology of appetite regulation, to the connections between mood, food and energy balance, and to the impact of different antidepressant treatments on appetite and body weight. The two figures are instrumental and clear and are useful in presenting the concepts detailed in the text. The table, resuming the effects of different classes of antidepressants on body weight, is clear and effectively summarizes the main message of the paper, including relevant literature on the topic. The text is complete, interesting, and pleasant to read. I have no specific concerns.Author Response
Please see the attachment.

Round 2
Reviewer 1 Report
Comments and Suggestions for Authors
You have responded well to my comments and made the appropriate changes accordingly.